# The Acquisition of Primary Amines from Alcohols through Reductive Amination over Heterogeneous Catalysts

Hao Huang [1], Yuejun Wei [2], Yuran Cheng [1], Shuwen Xiao [1], Mingchih Chen [2] and Zuojun Wei [1,*]

[1]  Key Laboratory of Biomass Chemical Engineering of the Ministry of Education, College of Chemical and Biological Engineering, Zhejiang University, Hangzhou 310027, China; huanghao0510@zju.edu.cn (H.H.); 22028086@zju.edu.cn (Y.C.); xiaoshuwen@zju.edu.cn (S.X.)

[2]  Graduate School of Business Administration, Fu Jen Catholic University, No. 510, Zhongzhen Road, New Taipei City 24205, Taiwan; weiyuejun0909@163.com (Y.W.); 406088165@mail.fju.edu.tw (M.C.)

*  Correspondence: weizuojun@zju.edu.cn

**Abstract:** The synthesis of primary amines via the reductive amination of alcohols involves a hydrogen-borrowing or hydrogen-transfer mechanism, which consists of three main steps: alcohol hydroxyl dehydrogenation, carbonyl imidization, and imine hydrogenation. Heterogeneous catalysts are widely used for this reaction because of their high performance and amenability to separation and reuse. However, the efficiency of reductive amination is limited by the dehydrogenation step, which is severely affected by the competitive adsorption of $NH_3$. We hope to improve the efficiency of reductive amination by increasing dehydrogenation efficiency. Therefore, in this overview, we introduce the research progress of alcohol reductive amination reaction catalyzed by heterogeneous metal catalysts, focusing on methods of enhancing dehydrogenation efficiency by screening the metal component and the acidity/alkalinity of the support. Finally, we propose some new strategies for the preparation of catalysts from the perspective of overcoming the competitive adsorption of $NH_3$ and speculate on the design and synthesis of novel catalysts with high performance in the future.

**Keywords:** competitive adsorption; heterogeneous catalysts; hydrogen-borrowing; primary amine; reductive amination



## 1. Introduction

Recently, the field of the transformation of biomass derivatives has become a research hotspot, in which the amination of alcohols and carbonyls is becoming increasingly popular [1–9]. Amines are versatile products that have applications in pharmaceuticals, pesticides, rubbers, surfactants, crosslinkers, and curing agents. Some organic amines are also of biological and medical significance [10–12]. Among the various amines, primary amines are considered to be the most important class of organic amine compounds because of their derivatization characteristics. Primary amines are especially important as they can undergo various derivatization reactions. For instance, aniline is the foundation of the dye industry. Binary primary amines, such as ethylenediamine and 1,2-propylenediamine, have two amino groups that can polymerize and have widespread uses in the production of pesticides, coatings, chelating agents, insect repellents, soil amendments, and lubricants and as emulsifiers, antifreeze agents, and organic solvents [13,14]. Therefore, the study of primary amines is conducive to the development of a series of valuable products that are very important in theoretical research and industrial applications.

There are many methods for the synthesis of amines, such as the amination of halogenated hydrocarbons, the hydrogenation of nitrile compounds, the amination of olefins, the reduction of nitroaromatic compounds [15,16], and the reductive amination of aldehydes/ketones/alcohols [17–23]. With the continuous improvement of the petroleum industry and the production of biomass-based alcohol compounds, the reductive amination of alcohols is becoming the most promising and effective method because of its

abundant source materials, low cost, and simple process [24–26]. Based on the reaction mechanism, the amination of alcohols can be divided into condensation amination and reductive amination.

Condensation amination uses dehydration catalysts like zeolite molecular sieves [27], metal oxides and their mixtures [28,29]. In 1909, Sabatier et al. first used ruthenium oxide as a catalyst to prepare a mixture of primary, secondary and tertiary amines from alcohol [30]. Since then, dehydration catalysts have been widely used in alcohol amination research and industrial production, e.g., for producing methylamines (methyleneamine, dimethylamine, trimethylamine, etc.) [31–33]. However, this method often requires a high reaction temperature that causes instability of alcohols and uncontrollable selectivity issues. Therefore, condensation amination is limited in low alcohol amination.

Reductive amination can be catalyzed by metal hydro-/dehydrogenation catalysts, which can be either homogeneous metal complex catalysts [34–38] or heterogeneous supported/unsupported metal catalysts [39–42]. Homogeneous metal catalysts are generally noble metals, such as Ru [43–47] and Ir [48,49], which also have drawbacks in product separation and reuse [50–53]. Some homogeneous catalysts also require co-catalysts such as acids, bases and salts [54,55]. Therefore, compared to the well-developed progress of homogeneous metal complexes with respect to the reductive amination of alcohols in the past few decades, heterogeneous metal catalysts are more green, economical, and reusable and thus worthy of being explored deeply [54].

## 2. Mechanism of the Reductive Amination of Alcohols

The mechanism of the alcohol reductive amination reaction, also known as the dehydroamination reaction [56], was first proposed by Schwoegler et al. in 1939 [57]. According to this "dehydrogenation-imidization-hydroamination" mechanism, an alcohol is first dehydrogenated over a hydrogenation/dehydrogenation catalyst to generate an aldehyde or ketone and then undergoes a nucleophilic attack by $NH_3$ and dehydration leading to the formation of an imine or Schiff base. Finally, the imine/Schiff base is reduced to a primary amine via catalytic hydrogenation. This viewpoint was supported by Bashkirov et al. [28], who studied the reductive amination of ethanol and identified the $\alpha$-hydrogen extraction in the dehydrogenation step as the rate-determining step. Figure 1 shows the reaction scheme, where Z is the active center on the catalyst, and * is the rate-determining step. Although the researchers agreed that the intermediates of the reductive amination of alcohol were aldehydes or ketones, the mechanism was not confirmed until 1983 when Baiker et al. [58], for the first time isolated the intermediate aldehydes from the amination reaction of alcohols.

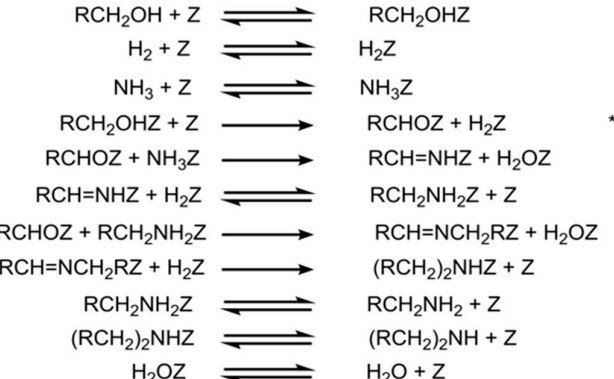

**Figure 1.** The mechanism of hydroamination of alcohols to primary amines. * is the rate-determining step [28].

In the reductive amination of alcohols, metals are usually used in reductive amination of alcohols as hydro-/dehydrogenation catalysts, such as homogeneous metal complex catalysts [59–63] and heterogeneous supported/unsupported metal catalysts [17,23,38,64–68].

According to the above reaction mechanism, exogenous $H_2$ is not necessary for the reductive amination process stoichiometrically because the first step of alcohol dehydrogenation can lend hydrogen to the third step of imine hydrogenation. This process is called the hydrogen-borrowing process or hydrogen transfer amination [54,55,69]. For example, Shimizu et al. [55] reported a supported metal catalyst, $Ni/Al_2O_3$, used to catalyze the reaction of over 10 kinds of alcohols with $NH_3$ to directly synthesize primary amines. For the reaction mechanism, they assumed that the alcohol was first dehydrogenated, thus yielding a carbonyl compound catalyzed by the Ni (0) active site. The carbonyl compound then reacts with $NH_3$ to form an imine, which finally reacts with the hydrogen transferred from Ni-H to form primary amines (Figure 2).

**Figure 2.** A plausible reaction mechanism of Ni-catalyzed amination of alcohols [55].

However, the reductive amination reaction is usually carried out under a certain $H_2$ pressure over heterogeneous catalysts, and these conditions are required to regenerate the active metal and remove the coke, maintaining the activity and selectivity of the catalysts [70–72]. Compared with the research on heterogeneous catalysis, hydrogen transfer amination is more common in homogeneous catalysis. Under the catalysis of the metal complex, the alcohol is dehydrogenated to form an active carbonyl compound. The removed "hydrogen" temporarily complexes with the catalyst, forming a hydride transition structure, which then returns the "hydrogen" to the intermediate imine and thus yields the target product amine. As early as 2008, Gunanathan et al. [73] used the Ru-PNP complex as a catalyst to successfully catalyze the reaction of primary alcohols with $NH_3$ to prepare primary amines. Thereafter, metal complexes such as Ru [17,36,59,60,67,74] and Ir [67,75,76] were mostly used to catalyze the hydrogenation transfer amination of alcohols. A significant advantage of the hydrogen transfer reaction is its high atomic economy, as water is the only by-product in the reaction [51,77,78].

Moreover, compared with organic amines, $NH_3$ has weaker nucleophilicity, which means that the reaction between alcohols and $NH_3$ leading to the formation of primary amines is more difficult than forming secondary or tertiary amines.

## 3. The Role of Heterogeneous Catalysts in the Reductive Amination of Alcohols

Heterogeneous metal catalysts have the advantages of relatively mild reaction conditions, high selectivity for primary amines, easy post-reaction treatment, and reusability. Therefore, they are increasingly being applied in the reductive amination of alcohols with $NH_3$ to prepare primary amines. At present, the metal active components are noble metals, such as Ru, Rh, Pt, and Pd, and non-noble metals, such as Co, Ni, Cu, and Fe, which are commonly used in reaction systems for the reductive amination of alcohols with $NH_3$ to produce primary amines. Table 1 classifies and compares the application of heterogeneous metal catalysts in the reductive amination of alcohols with $NH_3$ for producing primary amines in the past 30 years. The table provides important factors in the evaluation of heterogeneous metal catalysts, including catalyst types, model reactions, solvents, $NH_3$ and $H_2$ pressure, reaction temperature, reaction time, yield, and selectivity.

**Table 1.** Reductive amination of alcohols with ammonia to synthesize primary amines.

| Catalyst | Reaction | Solvent [a] | a/s [b] | Pressure (MPa) NH₃ [c] | H₂ | NH₃/H₂ [d] | T (°C) | t (h) | WHSV (h⁻¹) | Yield (%) | Select. (%) | Ref. |
|---|---|---|---|---|---|---|---|---|---|---|---|---|
| 58 wt% Ni/SiO₂ | (scheme) | - | 8.0 | - | - | 4.00 | 190 | - | 40.50 | 58.0 | 82.9 | [79] |
| 56 wt% Ni/SiO₂ | (scheme) | - | 60.0 | scNH₃ | - | 30.00 | 210 | - | 9.37 | 58.0 | 77.0 | [80] |
| Ni/SiO₂-AE | (scheme) | toluene | - | 0.3 | 0.4 | 0.75 | 160 | 2 | 71.7 | 76.9 | 86.0 | [81] |
| 17 wt% Ni/$\gamma$-Al₂O₃ | (scheme) | - | 4.0 | 0.012 | 0.018 | 0.67 | 170 | - | 4.29 | 62.1 | 69.0 | [70] |
| 10 wt% Ni/$\gamma$-Al₂O₃ | (scheme) | - | 3.0 | 0.009 | 0.018 | 0.50 | 190 | - | 0.55 | 28.4 | 35.0 | [82] |
| 5 wt% Ni/$\gamma$-Al₂O₃ | (scheme) | o-xylene | 2.2 | 0.400 | - | - | 160 | - | - | 81.0 | 97.6 | [55] |
| 17 wt% Ni/$\eta$-Al₂O₃ | (scheme) | - | 4.0 | 0.012 | 0.018 | 0.67 | 160 | - | 4.30 | 59.9 | 77.2 | [83] |
| 10 wt% Ni/CaSiO₃ | (scheme) | o-xylene | 2.2 | 0.400 | - | - | 160 | - | - | 86.0 | 90.5 | [69] |
| 24.3 wt% Ni/HZSM-5 | (scheme) | - | 2.0 | 0.040 | 0.040 | 1.00 | 170 | - | 25.00 | 61.2 | 68.0 | [84] |
| 10 wt% Co/SiO₂ | (scheme) | - | 2.0 | - | - | 5.00 | 180 | - | 0.25 | 30.2 | 56.0 | [85] |
| 9 wt% Co/SiO₂ | (scheme) | - | 2.0 | 0.007 | 0.030 | 0.23 | 180 | - | 1.00 | 32.8 | 49.5 | [86] |
| 23 wt% Co/$\gamma$-Al₂O₃ | (scheme) | - | 4.0 | 0.036 | 0.024 | 0.67 | 210 | - | 4.29 | 79.2 | 90.0 | [71] |
| 5wt% Co/$\gamma$-Al₂O₃ | (scheme) | tetrahydrofuran | 8.5 | 0.6 | 3 | 0.2 | 180 | - | - | 14.7 | 73.6 | [87] |
| 5 wt% Co/La₃O₄ | (scheme) | - | - | NH₃·H₂O | - | - | 160 | 6 | - | 68.6 | 89.4 | [68] |
| 3.5 wt% Cu/ZrO₂ | (scheme) | - | 5.0 | - | - | 1.00 | 250 | - | 1.80 | 42.8 | 63.0 | [88] |
| 19.8 wt% Cu/$\gamma$-Al₂O₃ | (scheme) | - | 200.0 | - | - | - | 420 | - | - | 25.4 | 26.2 | [89] |
| Fe | (scheme) | - | 3.0–10.0 | - | 3.00–5.00 | - | 230–250 | - | - | 35.0 | - | [90] |
| 5 wt% Ru/C | (scheme) | - | - | 0.400 | 0.200 | 2.00 | 150 | 20 | - | 83.8 | 84.6 | [91] |
| 5 wt% Ru/Al₂O₃ | (scheme) | tert-butanol | - | scNH₃ | 1.000 | 15.00 | 220 | 4.5 | - | 38.4 | 38.4 | [92] |
| 5.3 wt% Ru/C | (scheme) | - | 11.0 | NH₃·H₂O | 1.000 | - | 170 | 6 | - | 45.0 | - | [92] |
| 1Ru/TiO₂ | (scheme) | - | - | 0.0135 | 0.0985 | - | 200 | - | - | 60.0 | 85.0 | [17] |
| 5 wt% Ru/HBEA(25)_4 | (scheme) | - | 15.0 | - | 0.5 | - | 180 | 120 | - | 87.0 | 92.0 | [59] |
| 0.25 wt% Pt/$\gamma$-Al₂O₃ | (scheme) | - | 20.0 | - | - | 1.00 | 200 | - | 0.50 | 52.6 | 82.6 | [93] |
| 0.6 wt % Pd/MgO | (scheme) | toluene | - | 0.4 | - | - | 200 | 10 | - | 72.0 | 80.0 | [65] |
| 17 wt% Ni-Fe/Al₂O₃ | (scheme) | - | 8.0 | 0.024 | 0.012 | 2.00 | 150 | - | - | 79.3 | 83.0 | [94] |
| NiCuFeOₓ | (scheme) | xylene | 40.0 | 1.000 | - | - | 150 | - | - | 77.0 | - | [77] |
| Ni-Ce/$\gamma$-Al₂O₃ | (scheme) | - | 11 | 0.1 | - | 1.22 | 200 | - | 0.132 | - | - | [64] |
| NiCu/MgAlO | (scheme) | - | 5 | 0.1 | - | 0.76 | 200 | 6 | - | 97.6 | 98.9 | [38] |
| Ni₂Al-600 | (scheme) | tetrahydrofuran | - | 0.4 | - | - | 180 | 36 | - | 84.1 | 84.1 | [23] |
| 95 wt%Co-5 wt% Fe | (scheme) | - | 40.0 | scNH₃ | - | 30.00 | 195 | - | 6.92 | 32.8 | 36.0 | [95] |
| 95 wt%Co-5 wt% Fe | (scheme) | - | 60.0 | scNH₃ | - | 30.00 | 185 | - | 1.63 | 54.0 | 93.1 | [96] |
| 95 wt%Co-5 wt% Fe | (scheme) | - | 60.0 | scNH₃ | - | 30.00 | 195 | - | 4.57 | 40.9 | 43.0 | [97] |
| 95 wt%Co-5 wt% Fe | (scheme) | - | 60.0 | scNH₃ | - | 30.00 | 170 | - | 20.90 | 26.3 | 47.0 | [98] |
| Co₉₈.₅Ag₁.₅/$\gamma$-Al₂O₃ | (scheme) | tetrahydrofuran | 8.5 | 0.6 | 3 | 0.2 | 180 | - | - | 14.5 | 72.7 | [87] |
| Co-Ba-Fe/$\gamma$-Al₂O₃ | (scheme) | toluene | - | - | 5.000 | - | 165 | - | - | 53.4 | 73.3 | [99] |
| Ni-0.3Fe/Al₂O₃ | (scheme) | - | 8 | 0.024 | 0.012 | 2 | 150 | 0.5 | - | 79.3 | 83.0 | [94] |
| NiCu/MgAlO | (scheme) | - | 12 | - | - | - | 170 | 18 | 0.30 | 94.1 | 95.5 | [100] |
| 10 wt% Rh-In(1:1)/C | (scheme) | - | 10.0 | NH₃·H₂O | 5.000 | - | 180 | 24 | - | 9.5 | 89.0 | [101] |
| 30 wt% Rh-In(1:1)/C | (scheme) | - | 10.0 | NH₃·H₂O | 5.000 | - | 180 | 24 | - | 90.0 | 90.0 | [102] |

[a] "/" represents "gas–solid phase reaction or gas phase reaction, solvent-free", [b] "a/s" represents "molar ratio of ammonia to substrate alcohol", [c] "scNH₃" represents "supercritical ammonia", and [d] "NH₃/H₂" represents "molar ratio of NH₃ to H₂".

### 3.1. Monometallic Catalysts

In catalytic hydrogenation/dehydrogenation reactions, the H-H bond or O-H bond is generally dissociated and adsorbed on the surface of the metal active site, allowing subsequent reactions to proceed. The activity of monometallic catalysts for the reductive amination of alcohols to primary amines is related to the relevant supports, which can be divided into three categories: amphoteric supports (such as $\gamma$-$Al_2O_3$, $Y_2O_3$, $CeO_2$, $ZrO_2$, and ZnO), acidic supports (such as $TiO_2$ and $Nb_2O_5$), and alkaline supports (such as MgO and CaO). During the reductive amination reaction of alcohol, basic sites contribute to the extraction of alcohol protons, thereby increasing the rate of dehydrogenation, which is the rate-determining step of the reaction, while acidic sites are associated with hydrogen transfer. According to the reaction mechanism of alcohol reductive amination elaborated in Section 2 above, the first step of alcohol dehydrogenation can provide hydrogen for the third step of imine hydrogenation. Therefore, the availability of suitable acidic sites is conducive to the transfer of hydrogen, thereby increasing the yield of the primary amine product by increasing the rate of the imine hydrogenation reaction in the third step.

### 3.1.1. Ni-Based Catalysts

Ni-based catalysts are most commonly used because of their good activity and high selectivity for primary amines in the reductive amination of alcohols.

As early as 1990, Baiker et al. [79] reported the reductive amination of 1-methoxy-2-propanol with $NH_3$ over $Ni/SiO_2$ catalyst. They studied the effects of reaction parameters such as the reaction temperature, the molar ratio of substrate to $H_2$ and $NH_3$, and the mass space velocity on the performance of the catalyst. The conversion of 1-methoxy-2-propanol reached 70%, and the selectivity for the primary amine product 2-amino-1-methoxypropane reached 82.9% under the conditions of a reaction temperature of 190 °C, a molar ratio of 1-methoxy-2-propanol/$NH_3$/$H_2$ of 1/8/2, and a space velocity of 40.5 $g \cdot g^{-1} \cdot h^{-1}$. Additionally, the catalyst was gradually deactivated when there was no $H_2$ in the reaction system and recovered when $H_2$ was re-introduced.

In order to study the effect of Ni loading on catalyst performance, Shin et al. [70] prepared five different loadings (4–27 wt%) of $Ni/\gamma$-$Al_2O_3$ using an equal volume impregnation method for the reductive amination of isopropanol with $NH_3$ to isopropylamine. The results showed that when the Ni loading was 17 wt%, the metallic Ni had a better dispersion on the $\gamma$-$Al_2O_3$ support, and the metal had the highest reduction degree. Similarly, excess $H_2$ could effectively prevent the catalyst from forming a phase transition of metal nitrides inhibiting the catalyst's deactivation.

Shimizu et al. [55] applied heterogeneous metal catalysts in the hydrogen transfer amination of 2-octanol with $NH_3$ to prepare primary amines. By comparing the different metals on the $\gamma$-$Al_2O_3$ support, it was found that Ni had better activity for catalyzing hydrogen transfer amination than Cu, Co, and other noble metals (Pt, Re, Au, etc.). In addition, they further studied the catalytic hydrogen transfer amination performance of different supported Ni catalysts and that of Ni powder and Raney Ni. They found that the activity of the Ni catalysts was related to the supports and attributed this to the different acid–base sites on the supports, in which the basic sites of alumina could facilitate the extraction of protons from alcohols to promote the dehydrogenation step, and the acidic sites could be related to the hydrogen transfer step. Afterwards, the research group also prepared a $Ni/CaSiO_3$ catalyst using the ion exchange method, which has been successfully used in the hydrogen transfer amination of a variety of fatty alcohols and aromatic alcohols to generate primary amines [69].

Dumon et al. [103] compared the catalytic performance of $Ni/Al_2O_3$ and $Pd/Al_2O_3$ with regard to the reductive amination of n-butanol. The catalytic activity of Ni was predicted to be lower than that of Pd according to DFT (Density functional theory) calculations, with the corresponding energetic spans for the amination catalytic cycle being 1.22 eV and 1.13 eV, and these results contradicted the experimental findings indicating that Ni had a much stronger ability to catalyze alcohol dehydrogenation than Pd. Further DFT

calculations revealed that the spectator $NH_3$ adsorbates could promote the catalytic activity of the Ni catalyst: on the one hand, $NH_3$ can promote the rupture of the O-H bond and the formation of amine intermediates on the Ni surface; on the other hand, as shown in Figure 3, $NH_3$ can stabilize the imine on the surface of Pd through hydrogen bonding, which hinders the hydrogenation step of the imine.

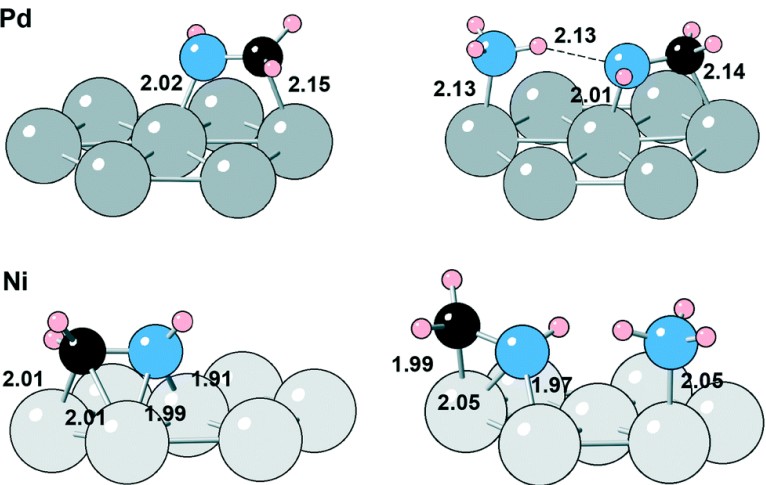

**Figure 3.** Imine adsorption in the absence and presence of $NH_3$ on Pd and Ni. The distances are given in Å. Reprinted with permission from Ref. [103]. Copyright 2018, Royal Society of Chemistry.

Our research group worked on the production of furfuryl alcohol and its derivative products [104,105]. In the reductive amination of furfuryl alcohol, we compared catalysts with different metal components (Ru, Rh, Pt, Pd, Co, and Ni) and found that Ni-based catalysts displayed the best performance, which was attributed to the difference in competitive adsorption of $NH_3$ and $H_2$ on different metal surfaces [106]. Next, in the reductive amination of 5-hydroxymethylfurfural, we conducted a DFT calculation of the difference in the adsorption energy of metal for $NH_3$ and $H_2$ and found that the difference in the adsorption energy of Ni for these two gases was lower than that of noble metals and Co, indicating that $NH_3$ will occupy less-active metal sites. This could be the reason why Ni-based catalysts show high performance in reductive amination [63].

Liu et al. [81] prepared a bifunctional Ni-based catalyst ($Ni/SiO_2$-AE) containing nickel nanoparticles and Lewis acid sites derived from nickel phyllosilicates via a modified ammonia evaporation method and used it in the reductive amination reaction of phenol. As shown in Figure 4, the $Ni/SiO_2$-AE catalyst enabled 89.4% phenol conversion and 86.0% selectivity for cyclohexylamine at 160 °C for 2 h, outperforming the catalysts formed via conventional deposition–precipitation ($Ni/SiO_2$-DP) and wetness-impregnation ($Ni/SiO_2$-WI). Based on the actual exposed metal surface area, the overturning frequencies (TOF) of $Ni/SiO_2$-AE, $Ni/SiO_2$-DP, and $Ni/SiO_2$-WI were calculated as 71.7, 49.0, and 8.7 ($h^{-1}$), respectively. Catalyst characterization results combining FTIR, XRD, TEM, and $NH_3$-TPD indicated that the nickel phyllosilicate structure was beneficial for the construction of small nickel nanoparticles (3 nm) and abundant Lewis acid sites necessary for phenol amination. The Ni nanoparticles exsoluted from nickel phyllosilicates participated in the activation of dihydrogen molecules, and the Lewis acid sites generated by the coordinated unsaturated $Ni^{2+}$ sites on the catalyst played a role in adsorbing and activating phenol. The synergistic effect of the dual functions enhanced the activity of the phenol amination reaction.

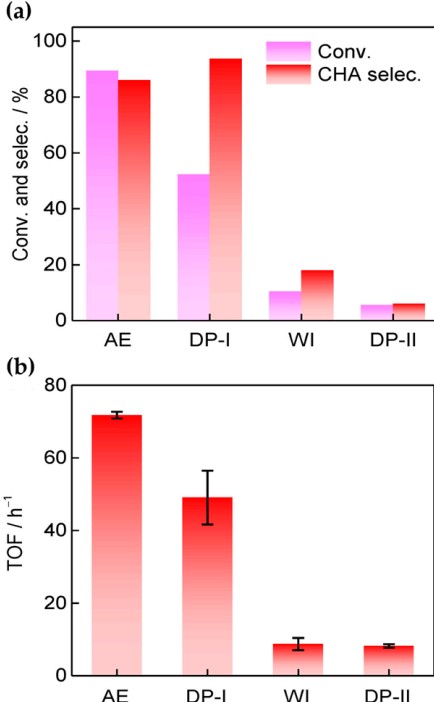

**Figure 4.** (**a**) The catalytic performance of catalysts for the amination of phenol to cyclohexamine (CHA). (**b**) TOF values for catalysts developed using different preparation methods with respect to phenol amination. Reprinted with permission from Ref [81]. Copyright 2023, Wiley-VCH GmbH.

From the above studies, it can be seen that temperature, $H_2$ pressure, $NH_3$ pressure, the structure of the substrate alcohol, catalysts, and other factors all have important effects on the catalytic reaction in the reductive amination of alcohols with $NH_3$ to produce primary amines. In the reductive amination of alcohols, a higher temperature is favorable for the dehydrogenation step, but an excessive temperature will cause side reactions of the substrate alcohol (such as cracking, dehydration, etc.), leading to a selectivity decrease for primary amines. Adding excess $NH_3$ to the reaction system can inhibit the formation of secondary and tertiary amines and thus increase the selectivity of primary amines. In addition, although the reductive amination of alcohol does not consume $H_2$, its presence can prevent the formation of metal nitrides and thus prevent the catalyst from deactivating. At the same time, the specific surface area, pore volume, and pH of the catalyst support will affect the activity and selectivity of the catalyst by affecting its dispersion and anti-sintering ability.

### 3.1.2. Co-Based Catalysts

Shin et al. [71] found that $Co/Al_2O_3$ had good activity and selectivity for the reaction of isopropanol with $NH_3$ that leads to the production of isopropylamine. With the increase in the Co loading, the conversion of isopropanol and the yield of isopropylamine increased significantly until the Co loading reached 23%. Additionally, with the increase in $NH_3$ partial pressure, the conversion of isopropanol was basically constant, the selectivity for isopropylamine gradually increased, and the selectivity for the dehydration by-product acetone decreased (Figure 5). This is because excess $NH_3$ promotes the reductive amination of acetone, which increases selectivity for isopropylamine.

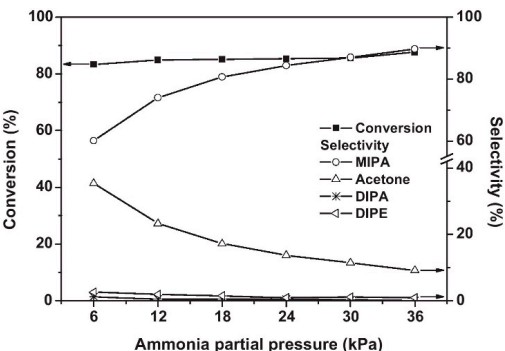

**Figure 5.** Influence of ammonia partial pressure on the reductive amination of 2-propanol over Co(23)Al$_2$O$_3$. Reaction conditions: T = 210 °C, WHSV (h$^{-1}$) = 4.29; feed composition of 2-propanol/H$_2$ (mol %) = 1/8. Reprinted with permission from Ref. [71]. Copyright 2012, Elsevier B.V.

Lei et al. [87] prepared a Co catalyst supported on γ-Al$_2$O$_3$ for the liquid-phase ammonolysis of ethylene glycol to produce high-value-added ethanolamine and explored the effect of different metal Co particle sizes on the selectivity of ethanolamine in the reductive amination of ethylene glycol. Among the sizes analyzed, Co nanoparticles above 4 nm showed the highest ethanolamine selectivity (73.6%) and the highest ammonolysis rate based on the total Co content (Figure 6a). Smaller Co nanoparticles (above 2 nm) were not only intrinsically less active in the ammonolysis of ethylene glycol but also exhibited a higher selectivity for glycolaldehyde due to an insufficient ability to catalyze the condensation reaction between glycolaldehyde and NH$_3$.

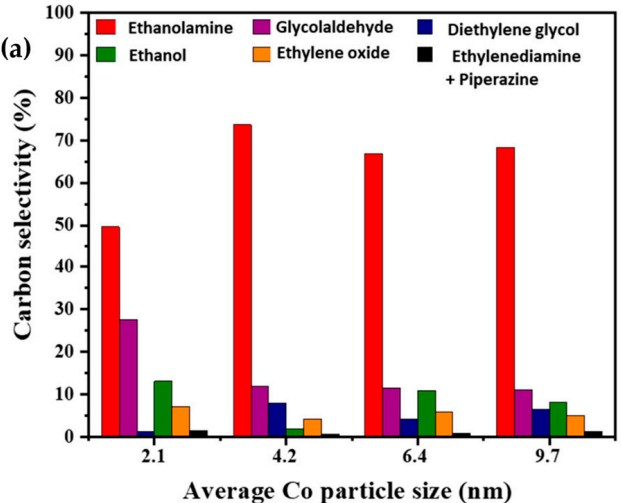
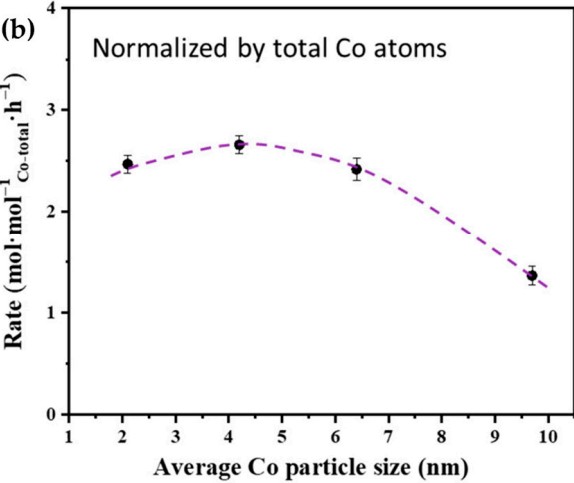

**Figure 6.** (**a**) Effects of Co particle size on the carbon selectivity of ammonolysis of ethylene glycol on Co/γ-Al$_2$O$_3$ catalysts. Red represents ethanolamine; purple represents glycolaldehyde; blue represents diethylene glycol; green represents ethanol; orange represents ethylene oxide; black represents ethylenediamine + piperazine. (**b**) Effects of Co particle size on the rate of ethylene glycol conversion over Co/γ-Al$_2$O$_3$ catalysts normalized by the total Co atoms. The dot represents the rate of ethylene glycol conversion. Reprinted with permission from Ref. [87]. Copyright 2021, MDPI.

### 3.1.3. Cu/Fe-Based Catalysts

Chary et al. [88] employed a constant volume impregnation method to prepare a series of Cu/ZrO$_2$ catalysts with different loadings (1–15 wt%) for catalyzing the reductive amination of cyclohexanol. Firstly, 2.5 wt% loading Cu was supported on ZrO$_2$, Nb$_2$O$_5$, and TiO$_2$ to screen for a suitable support. The highest yield of 57% cyclohexylamine was obtained over the catalyst supported on ZrO$_2$, which contained acid, alkali, reduction, and oxidation sites at the same time. The catalyst with a Cu loading of 3.5 wt% had the highest dispersion and the best catalytic performance. Moreover, the acidity and basicity of the

catalysts were studied by catalyzing the dehydrogenation of cyclohexanol and contrasted with the results of $NH_3$-TPD. It was discovered that acidic sites of moderate strength were favorable for the dehydration of cyclohexanol, but acidic sites of weak strength were beneficial for the dehydrogenation of cyclohexanol. Additionally, in their previous study [107], it was found that with the increase in the Cu loading, the alkalinity of the catalyst increased and the acidity of the catalyst decreased, and both of them tended to be stable with a higher Cu loading (more than 5.0 wt%).

Zhang et al. [89] studied catalysts supported on $\gamma$-$Al_2O_3$ to catalyze the reaction between ethanol and $NH_3$ and found that the selectivity for ethylamine in the product was not high when incorporating the metal component Cu. Furthermore, when the $NH_3$/ethanol molar ratio was low, aldehyde and acetonitrile accounted for the majority of the byproducts.

In early research, Kliger et al. [90] studied the mechanism of the reductive amination of alcohols using metal Fe catalysts and proposed an analogous hydrogen transfer mechanism.

For the reductive amination of alcohols, Cu and Fe are not usually used as active components alone but instead are generally used together with Co, Ni, and other metals as bimetallic/multimetallic catalysts.

### 3.1.4. Noble Metal-Based Catalysts

Noble metals are widely used in the reaction of aldehydes/ketones/alcohols with organic amines to produce secondary or tertiary amines due to their excellent hydro-/dehydrogenation activity. However, they are rarely used in the preparation of primary amines via the reductive amination of aldehydes/ketones/alcohols with $NH_3$. In a series of studies on the reductive amination of isophorone nitrile, furfuryl alcohol, and 5-hydroxymethylfurfural, our research group found that noble metal catalysts had no advantages over Ni and Co in terms of activity and selectivity [4,7,108,109]. For example, in the reductive amination of furfuryl alcohol and 5-hydroxymethylalcohol, we found that the yield of primary amine was very low when using Ru, Rh, Pt, and Pd as metal active components, which could be attributed to the competitive adsorption of hydrogen and $NH_3$ on the metal surface [63,106].

Among the catalysts for the reductive amination of alcohols with $NH_3$, Ru is the most used noble metal. Rose et al. [92] employed Ru/C, Pd/C, Pt/C, and other commercial catalysts to catalyze the reductive amination of biomass diols (isomannitol, isosorbide, etc.) with ammonium hydroxide to produce monoamine or diamine. It was found that under the reaction conditions of 140–180 °C with water as a solvent, Pd/C was basically inactive, and Pt/C had low activity. Ru/C proved to be the most active catalyst: using this catalyst, the total yield of monoamine and diamine reached 45% under the reaction conditions of 1 MPa of $H_2$, 1 g of 25 wt% ammonium hydroxide, 170 °C, and 6 h.

Ruiz et al. [110] studied the catalytic performance of different noble metal catalysts (Ru/C, Pd/C, Ir/C, Pt/C, and Os/C) on the reductive amination of dodecanol with $NH_3$ to produce dodecylamine. Under the reaction conditions of 150 °C, 0.4 MPa of $NH_3$, 0.2 MPa of $H_2$, and 24 h, the yield of dodecylamine was 83.8% over the Ru/C catalyst; the results are shown in Figure 7c. The authors studied the effects of $H_2$ pressure and $NH_3$ pressure on the reaction. In the case of excess $NH_3$ (applied to suppress the formation of secondary amines), changing the pressure of $NH_3$ had no significant effect on the conversion of dodecanol and selectivity for dodecylamine. When $H_2$ was not introduced, the conversion of dodecanol and the selectivity for dodecylamine were relatively low, and the main by-product was dodeconitrile. After the introduction of $H_2$ into the reaction system, the conversion of dodecanol and the selectivity for dodecylamine both increased. However, when the $H_2$ pressure was too high, dodecanol was cracked, producing a large amount of undecane. Moreover, it was found that the deactivation of the Ru/C catalyst occurred due to surface coking. Ruiz et al. [67] also studied the direct amination of dodecanol with $NH_3$ and $H_2$ over the catalysts of Rh, Pt, Ir, Ru, Ni, Cu, and Co supported on $SiO_2$. The results showed that Ru and Ir catalysts had greater activity, while Ni and Co catalysts exhibited better selectivity in alcohol reductive amination.

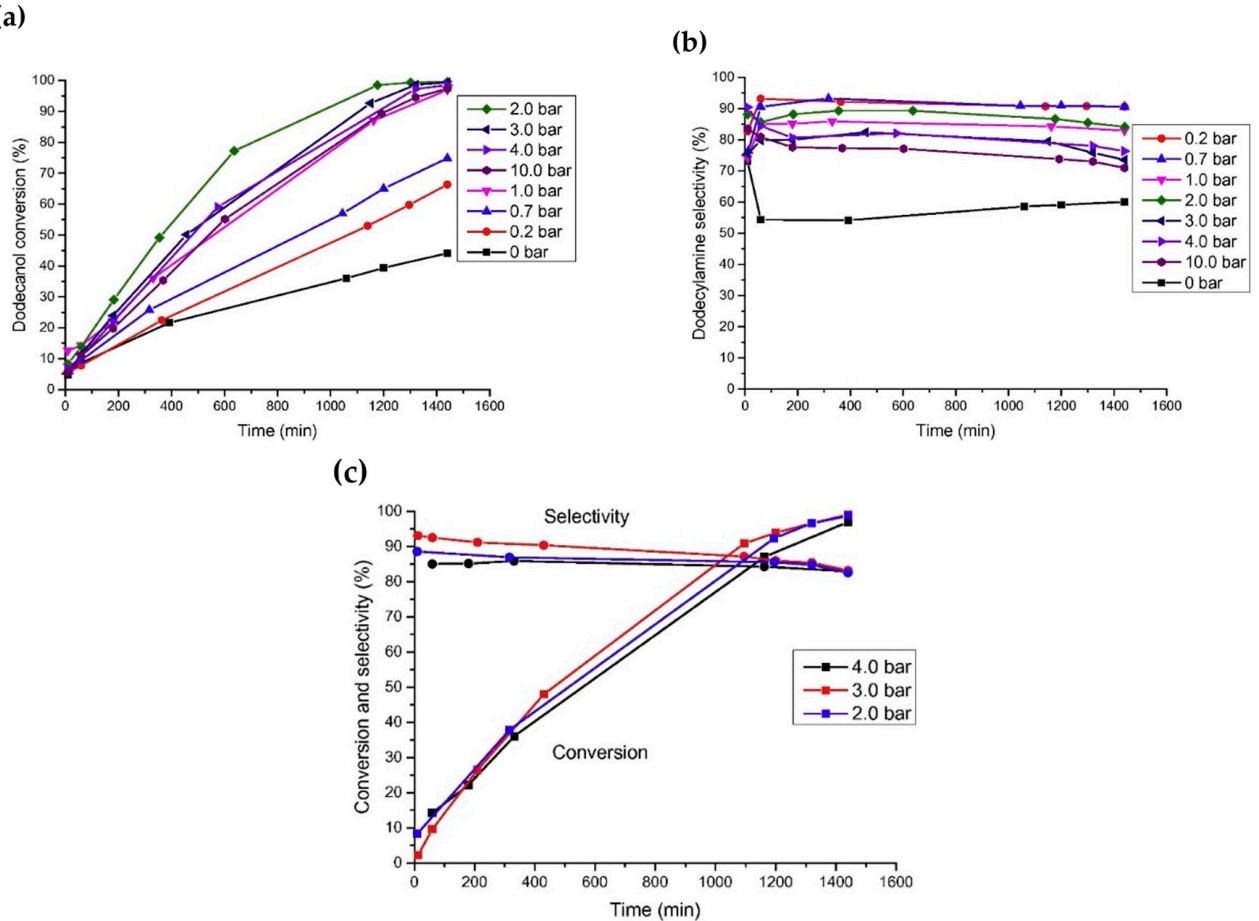

**Figure 7.** (**a**) Influence of hydrogen pressure on dodecanol conversion over Ru/C at 150 °C. (**b**) Influence of hydrogen pressure on selectivity toward dodecylamine over Ru/C at 150 °C. (**c**) Influence of ammonia pressure on dodecanol conversion and selectivity toward dodecylamine. Reprinted with permission from Ref. [110]. Copyright 2016, Elsevier B.V.

Fang et al. [59] investigated the reaction of 1-octanol with ammonia for the synthesis of 1-octylamine and examined the catalytic activity of loaded Ru catalysts on different acidic carriers including zeolites with various topologies and Si/Al ratios, in which the Ru/HBEA (Si/Al = 25) catalysts with 5 wt% Ru achieved a greater than 90% conversion of 1-octanol and 90% selectivity for 1-octylamine (Figure 8a). The authors attributed this result to the presence of moderately strong Brønsted/Lewis acid centers near the Ru nanoparticles located on the outer surface of HBEA.

## 3.2. Bimetallic/Multimetallic Catalysts

Compared with monometallic metal catalysts, bimetallic/multimetallic metal catalysts have the following main advantages in terms of their catalytic mechanisms:

1. Due to the addition of a second metal, the metals will interact with each other and have different electronic and chemical properties from the corresponding single metals, thus showing better catalytic activity, selectivity, and stability.
2. The addition of a second metal component can promote the reduction of the first metal precursor and improve the dispersion of metal particles on the support, strengthening the dehydrogenation site, thereby facilitating better catalytic activity and higher product selectivity.
3. The addition of a second metal can enable the formation of a highly dispersed bimetallic alloy on the nanostructure scale. The synergistic effect of a bimetallic alloy is as follows: the introduction of the second metal affects the electron cloud density of

the first metal, which is beneficial to the in situ reduction of the first metal, thereby improving catalyst activity.

4. The introduction of a second metal component can inhibit the competitive adsorption of $NH_3$ at the main metal catalytic sites, and more main metal catalytic sites can be used to catalyze the dehydrogenation reaction of alcohol, thus improving the dehydrogenation activity of the catalyst.

These properties are particularly important for catalytic processes that require high activity and high product selectivity [23,64,111].

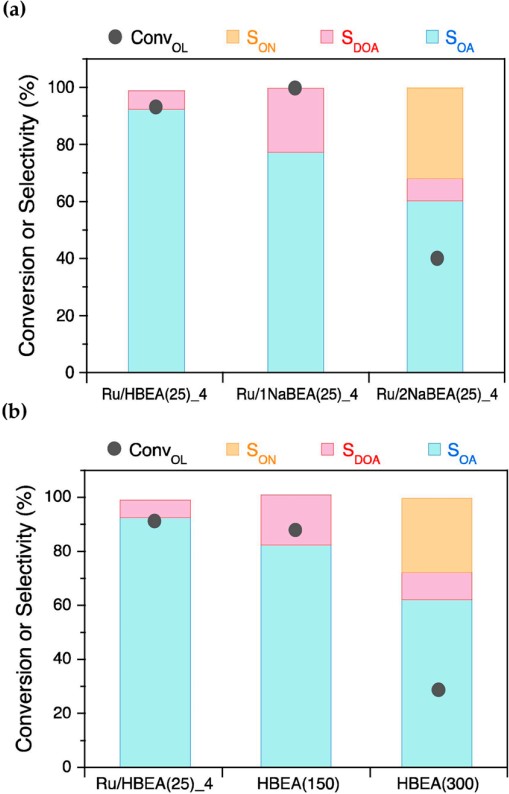

**Figure 8.** (**a**) Catalytic performance of Ru/BEA(25)_4 at different $H^+$-exchange degrees in the direct amination reaction of 1-octanol (OL) with $NH_3$. (**b**) Catalytic performance of Ru/BEA (x) as a function of Si/Al ratio (x) with x = 25, 150 and 300. Reprinted with permission from Ref. [59]. Copyright 2021, Elsevier B.V.

### 3.2.1. Ni-Based Catalysts

Cui et al. [77] prepared a Ni-based multimetallic catalyst, $NiCuFeO_x$, using a co-precipitation method, which was stable in air and was successfully used to catalyze the reductive amination of alcohols with $NH_3$ for producing primary and secondary amines. When studying the reaction between alcohol and $NH_3$, a superior yield of primary amines was obtained when ammonium carbonate was used as the $NH_3$ source. It is worth noting that the reaction was carried out under hydrogen-free conditions, which meant a hydrogen transfer mechanism was successfully achieved. The catalyst could be reused multiple times in the same reaction. This study provides a new idea for the use of the hydrogen transfer amination of alcohol with $NH_3$ to prepare primary amines over heterogeneous metal catalysts.

Shin et al. [94] studied the effect of metal loading and support morphology on the reductive amination of isopropanol with $NH_3$ and then further investigated the effect of the Ni/Fe molar ratio in Ni-Fe/$\gamma$-$Al_2O_3$ on the activity of the catalyst. They prepared 17 wt% Ni-Fe/$\gamma$-$Al_2O_3$ catalysts with different Fe/Ni molar ratios (0–0.7) through the same volume impregnation method. XRD characterization revealed that a Ni-Fe alloy had

formed in the reduced catalyst. $H_2$-TPR characterization showed that the reduction degree of Ni in the bimetallic catalyst significantly increased, and the reduction temperature was decreased compared to that of Ni/$\gamma$-Al$_2$O$_3$. Furthermore, XPS characterization showed that the content of Ni$^0$ in the catalyst changed with the change in the Ni/Fe molar ratio. When the Ni/Fe molar ratio was 0.3, the content of Ni$^0$ on the catalyst's surface reached a peak, and the activity and selectivity of the catalyst were the greatest, with no obvious deactivation after 100 h of reaction in the presence of hydrogen.

Our research group prepared a series of Ni-M/$\gamma$-Al$_2$O$_3$ (M = Mn, Re, Ce, Mo, Cr, Zr, Cu, Zn, La, and V) catalysts using the co-impregnation method for a 5-hydroxymethylfurfural "one-pot" reductive amination in order to produce 2,5-bis(aminomethyl)furan [112]. The results showed that the Mn-added catalyst displayed the best performance in the reaction. We considered that the addition of Mn endowed the Ni with higher surface dispersion and chemical stability through the dilution and electron transfer of Ni on the support surface.

Ma et al. [113] prepared Al$_2$O$_3$-supported Ni and Ni-Re catalysts using an impregnation method for the reductive amination of monoethanolamine. Compared with the Ni/Al$_2$O$_3$ catalyst, the Ni-Re/Al$_2$O$_3$ catalyst showed high activity and excellent stability and catalyzed the reaction to reach a superior ethanediamine selectivity. According to NH$_3$-TPD, CO$_2$-TPD, and PY-IR characterizations, the Ni-Re/Al$_2$O$_3$ catalyst exhibited a smaller Ni particle size (4–6 nm), more Ni$^0$ sites, and higher acid strength due to the introduction of Re. Moreover, the authors believed that the O atom in the -OH group was preferably adsorbed on the ReO$_x$ site during the amination of monoethanolamine due to its stronger oxygen affinity with Re compared to Ni. Compared with the bare Ni$^0$ surface, the dissociation energy of proton transfer and β-H elimination of monoethanolamine was significantly reduced on the Ni$^0$ surface modified by low-valence-state (<3) ReO$_x$. As a result, the adsorption structure on the ReO$_x$-Ni$^0$ surface facilitated the dehydrogenation process [114].

Chang et al. [115,116] added CeO$_2$ as a component to prepare Ni-CeO$_2$/Al$_2$O$_3$ to catalyze the synthesis of polypropylene diamine from polypropylene glycol. By increasing the content of Ce from 0 wt% to 2.5 wt%, 7.5% wt%, and 15 wt%, the total degrees of amine conversion increased from 64.5% to 66.8%, 71.1%, and 74.4% respectively, indicating that the catalytic performance was enhanced by the addition of CeO$_2$. This result may be attributed to the formation of a Ni-Ce-O interface on the nanostructure scale, providing a synergistic approach to promoting polypropylene diamine desorption (Figure 9).

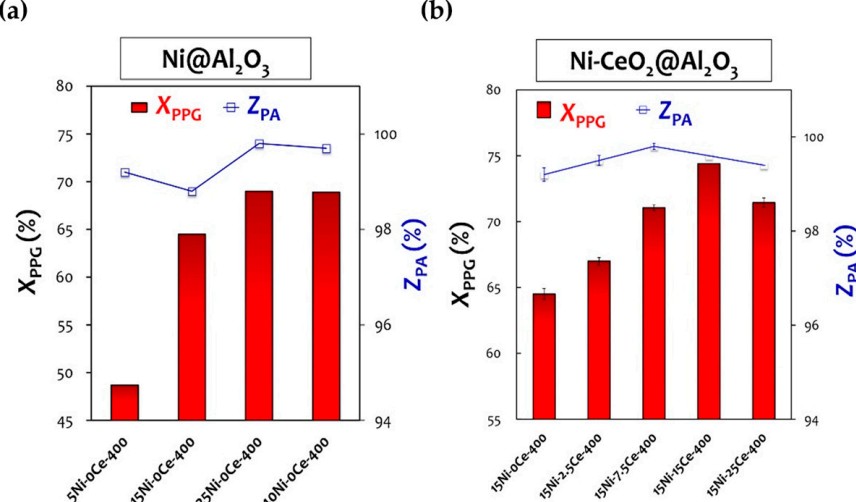

**Figure 9.** The conversion to total amine compound ratio ($X_{PPG}$) and the selectivity toward primary amine ($Z_{PA}$) using Ni@Al$_2$O$_3$ (**a**) and Ni-CeO$_2$@Al$_2$O$_3$ samples (**b**). Reprinted with permission from Ref. [115]. Copyright 2019, Elsevier B.V.

Then, Chang et al. [116] used Ni-CeO$_2$/Al$_2$O$_3$ to achieve a 99% selectivity and a 77% yield of polypropylene diamine. As shown in the SEM characterization presented in Figure 10, Ni and Ce had a uniform distribution in the mixed nanostructures, which meant that the Ni-Ce-O interface had been successfully generated. They believed that such oxygen vacancies at the Ni-Ce-O interface could promote the adsorption of the substrate and thus resist the competitive adsorption of NH$_3$ on the Ni surface. The Ni-Ce-O interface also provided additional active sites for the dehydrogenation of polypropylene glycol, shortening the reaction time. Hardly any Ni$_3$N crystals were found via characterization, suggesting that the addition of CeO$_2$ also inhibited the formation of nitrides on the surface of the catalyst. Furthermore, the addition of CeO$_2$ increased the acidity of the Ni-based catalysts, and the acidity increment could improve the stability of the Ni catalyst materials and weaken the NH$_3$ adsorption at the Ni-Ce-O interface, thus preventing the irreversible adsorption of NH$_3$/amines throughout the catalytic reaction process.

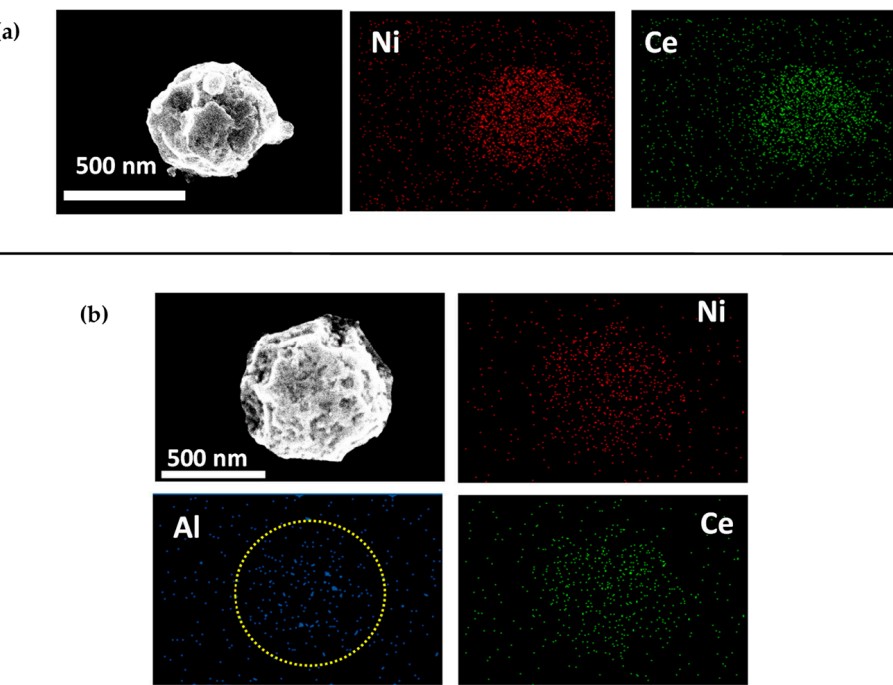

**Figure 10.** Representative SEM images with EDS elemental mapping. (**a**) 1Ni-0.5Ce-0Al. (**b**) 1Ni-0.5Ce-5Al. The yellow circle indicated that the mesoporous structure of 1Ni-0.5Ce-5Al was observed, indicating a successful formation of Al$_2$O$_3$ nanoparticle cluster (NPC) as support material of Ni-CeO$_2$ hybrid nanoparticle (NP) via gas-phase evaporation-induced self-assembly. Reprinted with permission from Ref. [116]. Copyright 2019, The Society of Powder Technology Japan.

Wei et al. [38] employed a non-precious-metal catalyst, namely, NiCu/MgAlO, for the reductive amination reaction of ethanol to prepare ethylamine. As shown in Figure 11d, the bimetallic 9%Ni-3%Cu/MgAlO catalyst achieved 98.7% ethanol conversion and high selectivity (98.9%) for ethylamines under optimal reaction conditions. The characterization results combining XRD, XPS, H$_2$-TPR, SEM, TEM, BET, and other catalyst characterization tools indicated that highly dispersed active Ni and Cu nanoparticles and a uniform particle size played significant roles in achieving the excellent catalytic performance and stability of the NiCu/MgAlO catalysts.

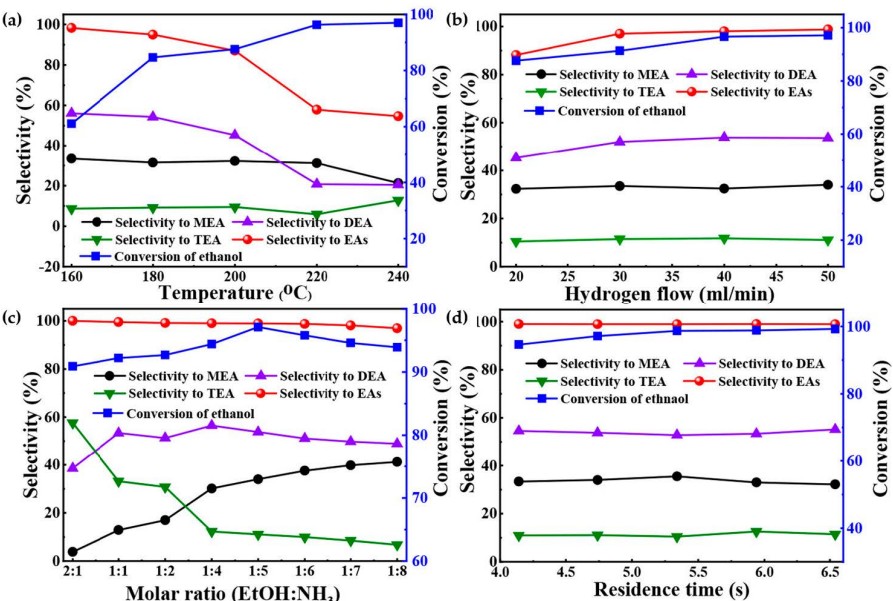

**Figure 11.** Amination results of ethanol with $NH_3$ over 9% Ni-3% Cu/MgAlO under different reaction conditions. (**a**) Molar ratio of ethanol to $NH_3$ is 1:5, flow rate of $H_2$ is 20 mL/min, flow rate of $NH_3$ is 38 mL/min, residence time is 5.34 s, and time on stream is 6 h. (**b**) Temperature is 200 °C, molar ratio of ethanol to $NH_3$ is 1:5, flow rate of $NH_3$ is 38 mL/min, residence time is 5.34 s, and time on stream is 6 h. (**c**) Temperature is 200 °C, flow rate of $H_2$ is 50 mL/min, flow rate of $NH_3$ is 38 mL/min, residence time is 5.34 s, and time on stream is 6 h. (**d**) Temperature is 200 °C, molar ratio of ethanol to $NH_3$ is 1:5, flow rate of $H_2$ is 50 mL/min, flow rate of $NH_3$ is 38 mL/min, and time on stream is 6 h. Reprinted with permission from Ref. [38]. Copyright 2023, American Chemical Society.

### 3.2.2. Co-Based Catalysts

Fischer et al. [97] researched the reductive amination of 1,3-propanediol over an Fe-modified Co catalyst in a supercritical $NH_3$ system. Firstly, through the study of the distribution of reaction products, it was found that the use of a supercritical $NH_3$ reaction system significantly improved the selectivity of primary amine products while inhibiting side reactions such as polymerization and cracking. Secondly, the effect of the promoter, Fe, was also researched. It was found that compared to the monometallic Co catalyst, not only was the stability of the Co-Fe catalyst better, but the Fe could also maintain the Co in a metastable β-Co of fcc crystal form in the catalyst, which was significantly more active than the α-Co of hcp form in the monometallic Co catalyst. Thirdly, the amount of added Fe also had an effect on the reaction. When the amount of Fe was more than 5%, a Co-Fe alloy was formed to decrease the catalytic activity. Fourthly, NaOAc and $(NH_4)_2HPO_4$ were used to treat the Co-Fe catalyst to study the effect of acidity and alkalinity on the performance of the catalyst. The results indicated that regardless of whether the acidity or basicity of the metal component changed, the performance of the catalyst would be reduced, and this effect could be attributed to the increasing content of α-Co and the formation of metal phosphates induced by the modification.

Zhao et al. [99] further synthesized a series of modified Co-Ba/γ-$Al_2O_3$ catalysts based on the catalysts used in the gas-phase reductive amination of 1,2-butanediol to 2-amino-1-butanol. Sr, Ni, Ca, Zn, La, Fe, Mg, Zr, Mn, and Cr metals were doped onto Co-Ba/γ-$Al_2O_3$ catalysts. Among them, Co-Ba-Fe/γ-$Al_2O_3$ showed the best catalytic performance. XRD, XPS, and TPR characterization demonstrated the formation of $Co_7Fe_3$ crystalline grains after the addition of Fe, which not only changed the catalytic performance of Co but also inhibited the formation of $CoAl_2O_4$, thus improving the stability and activity of the catalyst. Over the Co-Ba-Fe/γ-$Al_2O_3$ catalyst, the conversion of 1,2-butanediol maintained at 67.4–72.9% and the selectivity toward the target product 2-amino-1-butanol remained above 73.3% under the optimal reaction conditions for 100 h.

Yue et al. [68] proposed that Co has excellent dehydrogenation performance in the amination of 1,2-propanediol over Co/La₃O₄. However, the existence of La₃O₄ is a necessary condition for this reaction because Co alone cannot catalyze the reductive amination of 1,2-propanediol, which can be interpreted as a dehydrogenation process on the Co-La-O interface (Figure 12).

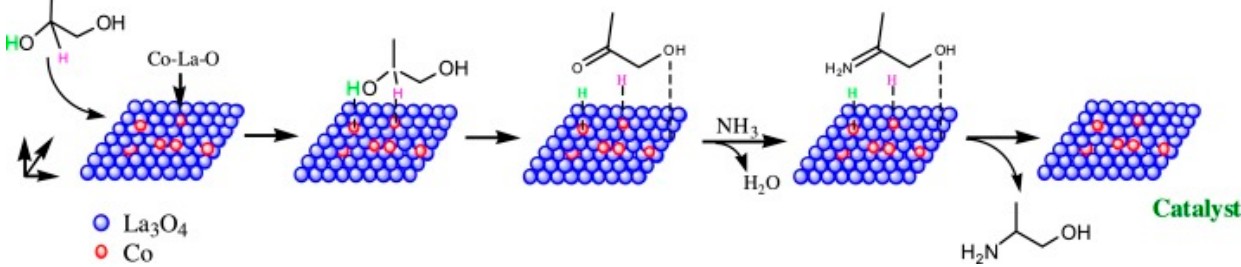

**Figure 12.** Schematic diagram of selective amination process for 1,2-propanediol. Reprinted with permission from Ref. [68]. Copyright 2019, Elsevier B.V.

Liu et al. [87] explored the effect of a second metal doping on a Co/γ-Al₂O₃ catalyst on the amination reaction of ethylene glycol to ethanolamine. As shown in Figure 13, the Co/γ-Al₂O₃ catalyst could be further improved by adding appropriate amounts of Cu, Ag, and Ru, while only in the case of Ag doping, the catalytic activity was further improved and a high chemical selectivity toward ethanolamine was maintained. Combined with kinetic and infrared spectroscopy evaluations, the results revealed that the dehydrogenation of ethylene glycol to glycolaldehyde was the rate-determining step in the reductive amination of ethylene glycol over cobalt-based catalysts. The introduction of Ag dopants weakened the adsorption of NH₃ on the Co surface. More Co sites could be used to catalyze the dehydrogenation of ethylene glycol, indicating that Ag doping can promote the ammonolysis activity of Co/γ-Al₂O₃ catalysts.

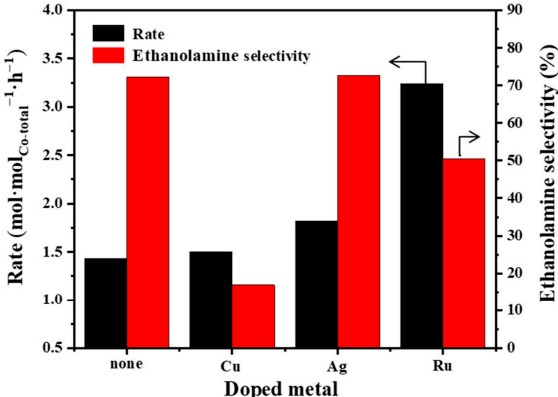

**Figure 13.** Effects of metal doping on the rate and selectivity of $Co_{98.5}M_{1.5}$/γ-Al₂O₃ (M = Cu, Ag, and Ru; 5 wt% Co) catalysts in ammonolysis of ethylene glycol (453 K; 0.6 MPa of NH₃; 3.0 MPa of H₂; 0.067 mol/L ethylene glycol in tetrahydrofuran solution; controlled at ~20% conversion). Reprinted with permission from Ref. [87]. Copyright 2021, MDPI.

### 3.2.3. Cu/Fe-Based Catalysts

Cui et al. [77] designed a cheap, non-noble-metal copper and nickel combination heterogeneous $NiCuFeO_x$ catalyst that showed excellent catalytic activity in the selective amination reaction of ammonia and alcohol and the N-alkylation reaction of primary amines. For the first time, several reactions, such as the selective amination of ammonia and alcohol for producing primary or secondary amines, the amination of aliphatic secondary alcohols, the amination of diols, and the amination of alcohols and dimethylamine, were realized. NiO and $Cu_2O$ were used as the main catalytic active species in the catalyst, with

FeO$_x$ serving as the support, and the catalyst was prepared through a Na$_2$CO$_3$ precipitation method at room temperature. These species were easily separated and reused due to their stable air and humidity properties and good magnetic properties. Using ammonia as the nitrogen source, the reductive amination reactions of four typical alcohols (benzyl alcohol, p-methoxybenzyl alcohol, 2-pyridinemethanol, and n-dodecyl alcohol) were verified in the absence of base and organic ligands. Moderate to good yields (59–77%) of the corresponding primary amines were obtained.

Hong et al. [94] successfully synthesized Ni-xFe/$\gamma$-Al$_2$O$_3$ catalysts with different Fe/Ni molar ratios (x = 0–0.7) via a co-impregnation method and investigated their catalytic properties in the reductive amination of isopropanol (IPA). The characterization results of the catalysts, obtained by combining N$_2$-adsorption, X-ray diffraction, H$_2$ temperature-programmed reduction, and X-ray photoelectron spectroscopy, showed that there was a synergistic effect of the bimetallic Ni-Fe on the $\gamma$-Al$_2$O$_3$ support. With the addition of Fe species, the surface Ni$^0$ content and the reducibility of the Ni component also increased. The maximum Ni$^0$ content was reached when the Fe/Ni molar ratio was 0.3. At the same time, the conversion rate and selectivity of the reductive amination of isopropanol to monoisopropylamine (MIPA) reached the highest levels. As shown in Figure 14, under optimized experimental conditions, the Ni-0.3Fe/Al$_2$O$_3$ catalyst achieved the highest IPA conversion (95.6%) and MIPA selectivity (83.0%), and it maintained relatively constant conversion and selectivity over 100 h on stream in a hydrogen atmosphere.

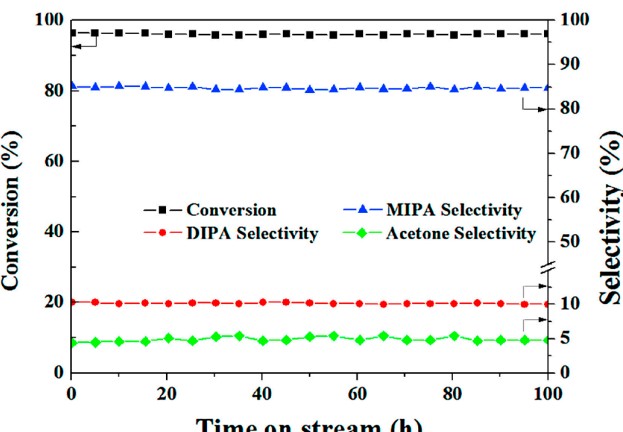

**Figure 14.** Long-term stability of the Ni-0.3Fe/Al$_2$O$_3$ catalyst in the reductive amination of isopropanol (IPA). Reaction conditions: T = 150 °C, SV = 300 mL min$^{-1}$ g$^{-1}$, and IPA/NH$_3$/H$_2$ molar feed composition = 1/8/4 with a constant IPA partial pressure of 3 kPa. Reprinted with permission from Ref. [94]. Copyright 2017, Elsevier B.V.

Wei et al. [100] reported a hydrotalcite-derived NiCu/MgAlO alloy catalyst that could catalyze the amination of cyclohexanol to prepare cyclohexylamine in high yields under atmospheric pressure. The results, presented in Figure 15, showed that the NiCu/MgAlO catalyst exhibited excellent catalytic performance and stability, and 98% cyclohexanol conversion with 95% selectivity toward cyclohexylamine was maintained after running for over 300 h at 170 °C. The super-high activity and stability of the NiCu/MgAlO catalyst were attributed to the high dispersion and uniform particle size (about 4 nm) of the active Ni−Cu alloy nanoparticles. DFT calculation results demonstrated that the Ni sites acted as the dehydrogenation active sites of cyclohexanol and that the Cu sites acted as the hydrogenation active sites of cyclohexylimine in the NiCu alloy catalyst. Furthermore, the authors also calculated the energy profiles for the potential comparative free energy of cyclohexanol with respect to cyclohexylamine on the NiCu alloy (111) facet, in which the first step of dehydrogenation and the last step of hydrogenation presented the highest energy barriers on the NiCu alloy (111) facet, indicating that these two steps are the main rate-determining steps in the alcohol amination process.

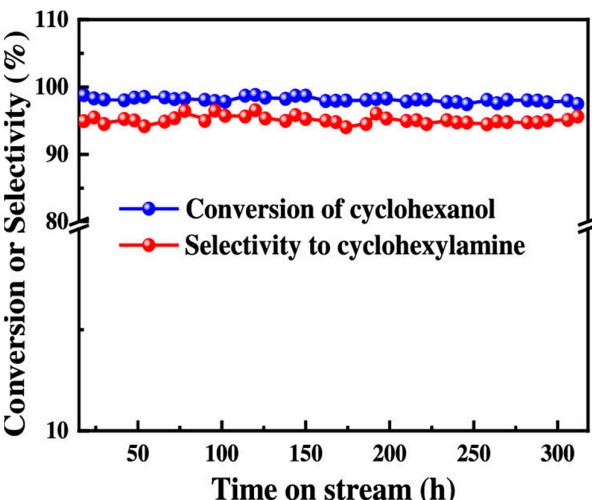

**Figure 15.** Durability test of the NiCu/MgAlO catalyst in the amination reaction. Reaction conditions: cyclohexanol/$NH_3$ = 1:12 (molar ratio), $H_2$ flow rate = 10 mL/min, $NH_3$ flow rate = 50 mL/min, temperature = 170 °C, residence time = 3.34 s, and GHSV = 1077 h$^{-1}$. Reprinted with permission from Ref. [100]. Copyright 2022, American Chemical Society.

### 3.2.4. Noble Metal-Based Catalysts

Takanashi et al. [101] prepared activated-carbon-supported Rh-based bimetallic catalysts with different metals (Ga, Sn, Zn, Ge, Bi, etc.) and used them for the reductive amination of 1,2-propanediol with $NH_3$ to amino alcohol in aqueous solvents under a 5.0 MPa $H_2$ atmosphere. They discovered that the activity of the Rh-In/C bimetallic catalyst for alcohol amination was significantly higher than that of several other metals, while the Rh/C and In/C monometallic catalysts had little activity when catalyzing the reaction, proving the synergistic effect of Rh-In on catalyzing alcohol amination. A further 1,2-propanediol dehydrogenation experiment indicated that the addition of the metal In significantly improved the efficiency of the alcohol dehydrogenation, especially in the presence of $NH_3$ or organic amines, and this improvement could be explained by the fact that the addition of In metal reduces the adsorption of the generated amino alcohol, $NH_3$, and carbonyl compound intermediates on the active components of the catalyst.

Takanashi et al. [102] then employed several catalysts with different In/Rh molar ratios for the reductive amination of 1,2-propanediol with $NH_3$ to amino alcohol in order to further determine the effect of the Rh-In/C catalyst structure. It was found that at a lower In/Rh ratio ($\leq$0.2), the Rh particles on the support surface formed fcc crystals with a particle size of less than 3 nm, and the particles were partially covered by In oxide. A tetragonal Rh-In alloy with larger particles (20 nm) was formed after adding more In; this alloy was in a crystal form following a transformation from a cubic Rh-In alloy. When the In/Rh molar ratio increased to 1, the TOF of the catalyst reached a peak, and the total selectivity toward the products 1-amino-2-propanol and 2-amino-1-propanol reached about 90%. However, the Rh-In alloy changed from tetragonal back to cubic when the In/Rh molar ratio continued to increase, causing a decrease in catalytic activity.

Tong et al. [117] studied the reductive amination of cyclopentanol to cyclopentylamine, in which a Pt-Co/CeO$_2$ catalyst had higher activity and selectivity than those of a Pt/CeO$_2$ catalyst. The research on two key catalytic steps (the dehydrogenation of alcohol and the over-hydrogenation of Schiff base intermediates) showed that the introduction of Co effectively promoted the dehydrogenation step and prevented Schiff bases from over-hydrogenating. The Co in the Pt-Co/CeO$_2$ catalyst was in the form of Co (II) (Figure 16). Notably, Co (II) could prevent a portion of the highly active Pt sites from being used for the over-hydrogenation of Schiff base intermediates, thereby inhibiting the formation of secondary amine by-products; furthermore, Co (II) could form a highly active interface with Pt sites for alcohol dehydrogenation, thus speeding up the rate-determining step.

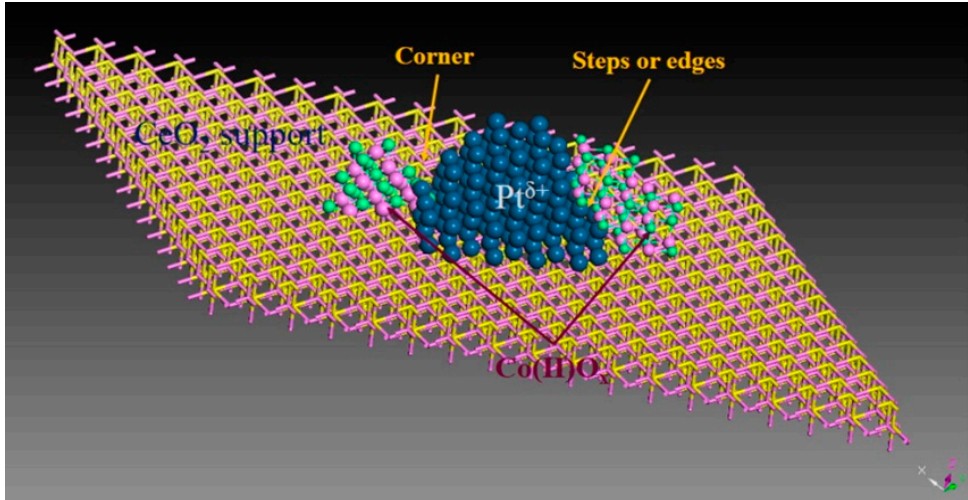

**Figure 16.** Geometric structure of 1%Pt-1%Co/CeO$_2$ catalyst. Pink: O, blue: Pt, green: Co, and yellow: Ce. Reprinted with permission from Ref. [117]. Copyright 2019, Elsevier Inc.

Ball et al. [118] found that the use of a Au-Pd/TiO$_2$ bimetallic catalyst for 1-hexanol amination can increase the conversion of 1-hexanol, which may be attributed to the geometric change of the Pd structure after being diluted in Au. From the FT-IR spectrum of adsorbed CO, the main CO binding pattern on the bimetallic catalyst indicated the existence of separated Pd surrounded by Au. The DFT calculation result of methanol amination with NH$_3$ on the Pd surface showed that when NH$_3$ was present, the imine intermediate would be adsorbed on the metal active sites and form hydrogen bonds with nearby NH$_3$, increasing the activation energy of the reaction and decreasing the overall reaction rate. After the addition of Au, NH$_3$ was less likely to form a hydrogen bond with the imine intermediate on the Pd sites due to the lower binding energy of NH$_3$ on the Au surface, increasing the conversion of hexanol.

Liu et al. [65] developed a sustainable method to replace traditional nitrobenzene-hydrogenation and halobenzene-amination in order to synthesize primary aniline. Cyclohexanone was used as an inducer, without additional hydrogenation in the reaction system, and the catalytic activity of supported Ru catalysts on different supports for the amination of phenol to aniline was investigated. Compared with catalysts such as Pd/Mg$_3$Al$_1$O, Pd/Mg$_1$Al$_1$O, and Pd/Al$_2$O$_3$, the Pd/MgO catalyst exhibited higher activity and selectivity toward primary aniline. Under the optimized reaction conditions, the conversion of phenol reached 90%, and the selectivity of the primary aniline was twice that obtained through phenol amination under a hydrogen atmosphere (Figure 17a). The characterization results indicated that the surface acid density of Pd/MgO was low and the adsorption capacity was weak; these aspects were beneficial to the desorption of aniline and reduced the condensation of aniline and cyclohexanone, thereby improving the selectivity of aniline. A method in which cyclohexanone was used as an inducer also realized the internal circulation of stoichiometric hydrogen species generated in situ from imine, promoting the development of the greener and more efficient synthesis of aniline compounds.

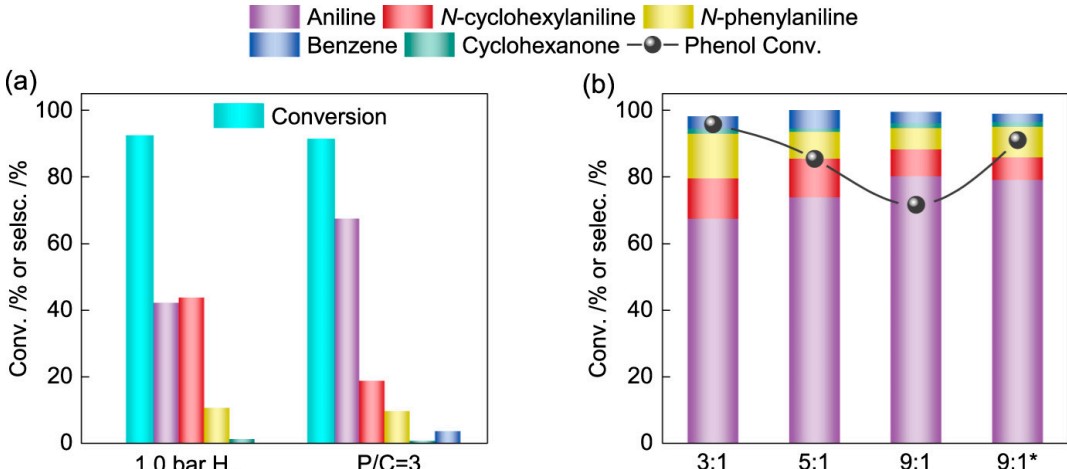

**Figure 17.** (**a**) Catalytic performance of 0.6 wt% Pd/MgO in phenol amination under different conditions and (**b**) catalytic performance of 0.6 wt% Pd/MgO under different molar ratios of phenol-to-cyclohexanone (P/C). Reaction conditions: 2.0 mmol substrate (phenol and cyclohexanone), 4.0 bar $NH_3$, 0.2 g catalyst, 20 mL toluene, and 600 rpm at 200 °C for 10 h. * represents the reaction after 20 h. Reprinted with permission from Ref. [65]. Copyright 2022, American Chemical Society.

Based on the results of the all the studies discussed in Section 3, in the preparation of primary amines, excess $NH_3$ is usually used to suppress the formation of secondary/tertiary amines and other by-products such as those produced by hydrocracking. However, noble metals have strong adsorption with respect to $NH_3$ and organic amines, and this inhibits their hydro-/dehydrogenation activity to some extent. Therefore, noble metals are not suitable for the preparation of primary amines with large amounts of $NH_3$. For the same reason, Ni and Co are superior to other metals in heterogeneous catalysts because they have weaker adsorption with respect to $NH_3$ and organic amines, and this maintains their hydro-/dehydrogenation activity to a certain degree. Additionally, under the same reaction conditions, the selectivity of noble metal catalysts toward primary amines is not high, so these catalysts are more often used in the reductive amination of alcohols with organic amines to prepare secondary and tertiary amines. At the same time, the effect of catalyst active components and supports on the selectivity of products is also important. When a second metal is added, the two metals will interact with each other, so they have different electronic and chemical characteristics from the corresponding single metal. The addition of the second metal component can not only promote the reduction degree of the first metal component and improve the distribution of metal particles on the catalyst surface, leading to better catalytic activity and higher product selectivity, but also improve the stability of the catalyst by suppressing the formation of metal nitrides, resisting the sintering of the metal components, and reducing the competitive adsorption of $NH_3$.

## 4. Conclusions and Outlook

The reaction mechanism of alcohol reductive amination to primary amines is a "hydrogen-borrowing mechanism": first, the alcohol hydroxyl undergoes a dehydrogenation reaction at the metal active sites on the catalyst, and the desorbed hydrogen is adsorbed on the metal; second, the dehydrogenated carbonyl group undergoes imidization with ammonia and then reacts with the hydrogen adsorbed on the metal to form a primary amino group. In the whole reaction, the dehydrogenation step, that is, the extraction process of $\alpha$-hydrogen, is the rate-determining step.

In the research on the preparation of primary amines via the reductive amination of alcohols, due to the importance of dehydrogenation/hydrogenation sites, heterogeneous metal catalysts have been widely used, which can be attributed to their excellent dehydrogenation/hydrogenation performance. In the design of catalysts, non-noble metals (such as Ni and Co) and noble metals (such as Pt, Ru, and Rh) are often selected as metal

active components, and $SiO_2$, $Al_2O_3$, activated carbon, etc., are often selected as supports. However, due to the presence of a rate-determining step, i.e., the dehydrogenation step, the results of the reductive amination of alcohols are often less than ideal. Although, in the reductive amination of aldehydes and ketones, a higher target product yield can be obtained through the action of a heterogeneous metal catalyst, when the same catalyst is applied to the reductive amination of alcohols, the product yield is often lower. Therefore, the use of heterogeneous metal catalysts for alcohol reductive amination to prepare primary amines is challenging to study.

Compared with other metals, Ni-based catalysts have obvious advantages in the preparation of primary amines via the reductive amination of alcohols. The reactions they catalyze usually have excellent substrate conversion and product selectivity. Our research results show that the competitive adsorption of ammonia on the surface of a metal active center will cause the hydrogen being removed by the hydroxyl group to lose its acceptor, which will reduce dehydrogenation activity and thus hinder the entire reductive amination reaction. However, Ni has the smallest adsorption energy difference between ammonia and hydrogen, which means it has the weakest competitive adsorption of ammonia. Therefore, Ni-based catalysts have higher catalytic performance for the reductive amination of alcohols to primary amines.

Since the dehydrogenation of hydroxyl groups limits the improvement of the efficiency of alcohol reductive amination, we hope to improve the dehydrogenation performance of catalysts. An effective method is adding a second metal component to the catalyst, which can not only strengthen the dehydrogenation sites by promoting the reduction degree of the first metal precursor and improving the dispersion of metal particles on the support but also promote desorption by reducing the dissociation energy of protons hydrogen. In addition, the second metal component can inhibit metal site poisoning caused by ammonia, thereby increasing the dehydrogenation activity of the catalyst. Another idea is controlling the acidity/alkalinity of the support. The basic site of the support can promote the reduction of metal components into active sites, resisting the competitive adsorption of ammonia on the metal sites to improve the dehydrogenation activity of the catalyst. In addition, the basic site itself can also promote desorption by reducing the dissociation energy of hydrogen protons. The acid sites of the support can increase the dispersion of metal on the support and can even be used as an additional hydrogen acceptor to improve the dehydrogenation activity of the catalyst. One of the keys to improving dehydrogenation efficiency in alcohol reductive amination is to reduce the competitive adsorption of ammonia. Although the regulation of the metal component and the acid and basic sites of the support can suppress this, unfortunately, there is no general catalyst that is very effective for most alcohol reductive amination reactions. Therefore, we hope to design a catalyst that fundamentally eliminates the competitive adsorption of ammonia, and we will think about preparing a catalyst with a cage structure, which can fix the metal active sites in the cage and block the ammonia from the cage, thereby suppressing the ammonia competitive adsorption. Given this, MOF (Metal organic framework) materials can achieve site isolation through the confinement effect, thereby stabilizing metal active sites. At the same time, MOF material catalysts also have high stability, recyclability, and reusability [119,120], so we will consider applying them in the reductive amination of alcohols.

**Author Contributions:** Conceptualization, Z.W.; methodology, Y.C., S.X. and Z.W.; investigation, H.H.; resources, Z.W.; data curation, H.H., Y.C. and S.X.; writing—original draft preparation, H.H.; writing—review and editing, H.H., Y.W., S.X., M.C. and Z.W.; visualization, H.H., Y.W., M.C. and S.X.; supervision, Y.W., S.X., M.C. and Z.W.; project administration, Z.W.; funding acquisition, Z.W. All authors have read and agreed to the published version of the manuscript.

**Funding:** This research was funded by the National Natural Science Foundation of China (21878269 and 21476211) and the Fundamental Research Funds for the Central Universities.

**Data Availability Statement:** Not applicable.

**Conflicts of Interest:** The authors declare no conflict of interest.

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
