# Peer review of "The Acquisition of Primary Amines from Alcohols through Reductive Amination over Heterogeneous Catalysts"

_catalysts, doi:10.3390/catal13101350_

Round 1

Reviewer 1 Report

 The authors have documented a good review of the reductive amination process.

The discussion of the different catalyst systems is good. It is, however, recommended to address the following comments to improve the quality of this article.

1.       Please provide a figure comparing the catalytic activity (conversion, and selectivity) of these various catalysts.

2.       It would be great if the authors could elaborate more on the mechanistic aspects of mono-metallic and multicomponent catalysts and provide future directions for improvement.

Author Response

Thank you very much for taking the time to review this manuscript. Based on your valuable review comments, we have addressed the comments one by one accordingly and listed the answers in the attached file. Please find the detailed responses below, as well as the corresponding revisions and corrections highlighted in the revised manuscript.

Reviewer 2 Report

Please, remember to quote also:

a) ACS Sustainable Chem. Eng. 2022, 10, 5526-5537 https://doi.org/10.1021/acssuschemeng.2c00115 

b) ACS Catal. 2020, 10, 3404−3414 https://dx.doi.org/10.1021/acscatal.9b05525 

Author Response

(The authors gave the same response as above.)

Reviewer 3 Report

The review article describes the available methods for synthesizing primary amines from alcohols and ammonia in presence of heterogeneous catalysts. The review article is well structured with the authors describing a clear introduction with reaction mechanism, currently employed monometallic and bi or multimetallic catalysts based on noble metals and non-noble metals. This work is surely a good contribution to the scientific community especially for those working in the area heterogenous catalysis, precisely for those work in the field of synthesis of primary amines by reductive amination strategy.

The authors clearly demonstrated the current works carried out in this area using various heterogenous catalysis. The authors were also able to fill the specific gap in this field by giving a future prospectus of this area of research. The references are adequate and the conclusions are clear leading to the opening of new opportunities in this field (especially the possible utilization of cage-structured catalysts and MOF materials). Overall, this review work is a significant contribution to the science and society and hence I recommend to accept this review article for publication in "Catalysts" after minor revision. 

Some of the specific comments that needs to be addressed are detailed below:

1) There are some instances where the authors have used badly worded phrases. Hence it is recommended to thoroughly proofread the article and correct the minor grammatical errors and restructure the complex sentences to present it in a more readable form.

2) In monometallic catalysts section, the authors have described the reports on Cu/Fe metal based catalysts. This is missing in multi or bimetallic catalysts section. I would like to know whether the authors couldn't find any reports on that direction which indicates that no one has carried out that research.

After addressing these comments, this review article may be published. 

Some minor English language editing is needed.

Author Response

(The authors gave the same response as above.)
